# Cyclic Nigerosyl-Nigerose as Oxygen Nanocarrier to Protect Cellular Models from Hypoxia/Reoxygenation Injury: Implications from an In Vitro Model

**DOI:** 10.3390/ijms22084208

**Published:** 2021-04-19

**Authors:** Claudia Penna, Saveria Femminò, Fabrizio Caldera, Alberto Rubin Pedrazzo, Claudio Cecone, Edoardo Alfì, Stefano Comità, Takanobu Higashiyama, Francesco Trotta, Pasquale Pagliaro, Roberta Cavalli

**Affiliations:** 1Department of Clinical and Biological Sciences, University of Turin, 10043 Turin, Italy; claudia.penna@unito.it (C.P.); saveria.femmino@unito.it (S.F.); edoardo.alfi@unito.it (E.A.); stefano.comita@unito.it (S.C.); 2Department of Medical Sciences, University of Turin, 10126 Turin, Italy; 3Department of Chemistry, University of Turin, 10125 Turin, Italy; fabrizio.caldera@unito.it (F.C.); alberto.rubinpedrazzo@unito.it (A.R.P.); claudio.cecone@unito.it (C.C.); 4Hayashibara CO., LTD./Nagase Group 675-1 Fujisaki, Naka-ku, Okayama 702-8006, Japan; takanobu.higashiyama@hb.nagase.co.jp; 5Department of Drug Science and Technology, University of Turin, 10125 Turin, Italy

**Keywords:** cyclic nigerosyl nigerose, oxygen delivery, myocardial infarction, ischemia, reperfusion

## Abstract

Heart failure (HF) prevalence is increasing among the aging population, and the mortality rate remains unacceptably high despite improvements in therapy. Myocardial ischemia (MI) and, consequently, ischemia/reperfusion injury (IRI), are frequently the basis of HF development. Therefore, cardioprotective strategies to limit IRI are mandatory. Nanocarriers have been proposed as alternative therapy for cardiovascular disease. Controlled reoxygenation may be a promising strategy. Novel nanocarriers, such as cyclic nigerosyl-nigerose (CNN), can be innovative tools for oxygen delivery in a controlled manner. In this study we analyzed new CNN-based formulations as oxygen nanocarriers (O_2_-CNN), and compared them with nitrogen CNN (N_2_-CNN). These different CNN-based formulations were tested using two cellular models, namely, cardiomyoblasts (H9c2), and endothelial (HMEC) cell lines, at different concentrations. The effects on the growth curve during normoxia (21% O_2_, 5% CO_2_ and 74% N_2_) and their protective effects during hypoxia (1% O_2_, 5% CO_2_ and 94% N_2_) and reoxygenation (21% O_2_, 5% CO_2_ and 74% N_2_) were studied. Neither O_2_-CNN nor N_2_-CNN has any effect on the growth curve during normoxia. However, O_2_-CNN applied before hypoxia induces a 15–30% reduction in cell mortality after hypoxia/re-oxygenation when compared to N_2_-CNN. O_2_-CNN showed a marked efficacy in controlled oxygenation, which suggests an interesting potential for the future medical application of soluble nanocarrier systems for MI treatment.

## 1. Introduction

The nanocarriers were studied for tissue-specific targeted delivery, triggered release, and co-delivery of synergistic drug combinations to develop new efficient therapeutics [1].

As a result of the high aerobic metabolism, the myocardium already exhibits a high extraction of oxygen in resting conditions compared to other muscle tissues. The heart is an organ with elevated consumption of oxygen in its basal condition, and under several different conditions, e.g., anemia, high altitude exposure, or vessel stenosis, cardiac oxygenation may become critical, leading to tissue hypoxia. In particular, during acute coronary occlusion, cardiac cell death occurs. This is the first phase of acute myocardial infarction (AMI) [2].

Complete and timely restoration of blood flow is mandatory in order to save the ischemic myocardium from irreversible ischemic injury; however, reperfusion induces additional injury, i.e., a *reperfusion injury*, and contributes to the final infarct size [3]. The lack of oxygen during ischemia and the subsequent re-introduction of oxygen during reperfusion, which lead to the massive formation of reactive oxygen species (ROS), are the basis of cardiac death and the inflammatory process, typical of ischemia/reperfusion injury. During AMI, a vicious cycle of deleterious events, including oxidative stress and inflammation, induces cardiac damage, impairing both diastolic and systolic function [2,4].

It is important to note the apparent paradox that ROS, which are normally considered lesion factors, at low concentrations and at the appropriate time of formation would induce myocardial protection [2,5,6,7,8]. Thus, the production of devices for the controlled delivery of oxygen can be of fundamental importance for reducing ROS lesions and favoring redox-induced protection.

Protective strategies, such as slow re-flow procedures, have been proposed to limit re-oxygenation-induced lesions. However, this procedure has been abandoned, as it presents the problem of stagnant blood and consequent white blood cell infiltration [2,9]. Moreover, an intermittent re-flow (post-conditioning mode) has been suggested as an alternative strategy. Furthermore, intermittent reperfusion has raised several concerns in clinical settings, and full reperfusion is still the treatment of choice [10]. The ability of nanoparticles to aid in drug delivery by targeting regions affected by organ ischemia has been studied by many authors [11,12,13,14]. However, to the best of our knowledge, only one study has tested a polymer nanoparticle approach for delivering oxygen in a controlled manner to limit I/R injury [15].

Cyclic nigerosyl-nigerose (CNN) is an interesting and innovative nanocarrier for anti-cancer drug delivery in the cross-linked forms previously shown [16]. The major advantage of CNN is that its natural structure is a reservoir for oxygen.

CNN is a non-reducing cyclic tetrasaccharide with an unusual structure, consisting of four d-glucopyranosyl molecules connected by alternate α-(1-3) and α-(1-6) glycosidic bonds (Figure 1). The presence of two oxydryl groups oriented to the cavity of the molecules leads to a more hydrophilic polarity of the CNN cavity in comparison with the well-known cyclodextrin one. The smaller cavity is much more prone to hosting small molecules of suitable size and polarity. 

The objectives of this study are to evaluate oxygen release kinetics and to ascertain whether different concentrations of CNN loaded with oxygen—or with nitrogen as a control—can modulate cell growth and reduce cell mortality in a hypoxia/re-oxygenation (H/R) protocol when given before or after hypoxia, using two different cellular models.

## 2. Results

### 2.1. In Vitro Oxygen Release Kinetics

The pH value of cyclic nigerosyl-nigerose solution, either in the presence or in the absence of oxygen, was 6.0, and the osmolarity was about 300 mOsm. Hemolysis testing was performed (see CNN samples at all the concentrations tested), indicating good biocompatibility and a tonicity value suitable for the cell experiments. The in vitro oxygen release kinetics from CNN solution (NaCl 0.9% *w*/*v*) in the two receiving phases, at 37 °C, are shown in Figure 2. As can be seen, O_2_ concentration reached a maximum level after 15 h both in saline solution and in a cell culture medium.

A sustained oxygen release was observed in both of the receiving phases, reaching an oxygen concentration of 7.08 mg/L and 5.15 mg/L, in the saline solution and the cell culture medium, respectively.

The in vitro release profile of oxygen from the oxygen-loaded CNN glycerol solution containing dextran-70 (6% *w*/*v*) was compared to that of a common plasma volume expander (5% *w*/*v* glucose solution containing dextran-70 (6% *w*/*v*)) saturated with oxygen (Figure 3).

The characteristics of the oxygen-loaded CNN samples are reported in Table 1.

### 2.2. Dose–Response and Growth Curve in Normoxic Conditions

We report in Figure 4 results obtained with different concentrations (0.2, 2, 10, and 20 μg/mL) of oxygenated (O_2_-CNN, upper panels) and non-oxygenated (N_2_-CNN, lower panels) nanocarriers, in normoxic conditions. The exposure of H9c2 to different concentrations of CNN nanocarriers for 2 hours did not affect cell vitality, regardless of the presence of oxygen.

To evaluate the effect of CNN on cellular growth, we have treated two different cell lines (H9c2 and HMEC) at a concentration of 10 μg/mL, for varying periods of time (24, 48, 72, 96, and 120 h). Both the H9c2 and HMEC proliferations were unaffected by either O_2_-CNN or N_2_-CNN given for 5 days at 10 μg/mL (Figure 5A,B).

### 2.3. Untreated Cells: Normoxic and H/R Conditions

Untreated (not exposed to CNN) H9c2 or HMEC cell lines were subjected to the H/R protocol, in order to evaluate their response to hypoxia and re-oxygenation. As can be seen in Figure 6A,B the untreated H9c2 at the end of the H/R protocol (CTRL H/R) displayed a 47% reduction in vitality compared to the CTRL group. Indeed, cell vitality was 100 ± 6% in CTRL and 63 ± 10% in CTRL H/R (*p* < 0.001 CTRL vs. CTRL H/R). As can also be seen in Figure 6C,D the untreated HMEC (not exposed to CNN) at the end of the H/R protocol (CTRL H/R) also displayed a 47% reduction in vitality compared to the CTRL group. Indeed, the cell vitality was 100 ± 2% in CTRL and 63 ± 1% in CTRL H/R (*p* < 0.001 CTRL vs. CTRL H/R).

### 2.4. CNN-Pre-Treated Cells in H/R Conditions

In order to evaluate whether O2-CNN presents protective effects with respect to the H/R protocol, we pre-treated both cell lines (H9c2 and HMEC) with increasing concentrations of the two formulations of CNN (0.2, 2, 10, and 20 μg/mL). Figure 6A,B shows that the H9c2 cell line was protected only by O_2_-CNN pre-treatment at high concentrations of 10 and 20 μg/mL: at these concentrations, vitality was significantly higher than in CTRL H/R conditions (*p* < 0.0001 for both concentrations). N_2_-CNN was not protective at any concentration: vitality was not significantly higher than in CTRL H/R conditions. Figure 6C,D shows that similar results were obtained for HMEC. In fact, the HMEC cell line was also protected by O_2_-CNN pre-treatment at high concentrations of 10 and 20 μg/mL: vitality was significantly higher than CTRL H/R (*p* < 0.01 and *p* < 0.0001, respectively). N_2_-CNN was not protective at any tested concentration.

### 2.5. CNN Post-Treated Cells in H/R Conditions

At the end of the hypoxic period, both cell lines were treated with O_2_ or N_2_-CNN at increasing concentrations (0.2, 2, 10, or 20 μg/mL). In Figure 7A,B, we show that the H9c2 cell line was protected by O_2_-CNN post-treatment only at the highest concentration of 20 μg/mL: vitality was significantly higher than CTRL H/R (*p* < 0.0001). Meanwhile, N_2_-CNN was not protective at any tested concentration; in fact, the mortality was similar to that in CTRL H/R conditions. Surprisingly, post-treatment with O_2_-CNN in the HMEC cell line was not protective at any concentration used, as shown in Figure 7C,D.

To corroborate the MTT-based data in terms of cell death and hypoxia mechanisms in cardiomyoblasts (H9c2), we performed Western blot analyses of cleaved caspase-3 and hypoxia-inducible factor 1-alpha (Hif-1α), respectively (Figure 8). These two assays were analyzed at O_2_-CNN concentrations that displayed protective results via MTT. Both assays displayed a trend in line with the results obtained via MTT. Indeed, Hif-1α and cleaved caspase-3 were both reduced by O_2_-CNN pre-treatment. However, in post-treatment, O_2_-CNN seemed less protective.

## 3. Discussion

This study strongly indicates that oxygenated CNN is beneficial during an H/R protocol, and limits cell death in two relevant cellular models (i.e., cardiomyocyte-like, H9c2, and endothelial, HMEC) when applied before the hypoxic insult. When applied after hypoxia, at the beginning of re-oxygenation, its protective properties are limited to high concentrations, and in the H9c2 model only. Moreover, our results demonstrate that neither O_2_-CNN nor N_2_-CNN has toxic effects, and they do not affect cell vitality or growth in normoxia, in two different cellular models. When given in the post-hypoxic phase, O_2_-CNN was protective only at high concentration, and only in one cellular model, underlining the different susceptibility to re-oxygenation injury of the cardiac and the endothelial cell lines.

Despite the success and recent improvement of interventional coronary reperfusion strategies, morbidity and mortality from AMI are still considerable. Myocardial infarct size is a major determinant of prognosis in heart failure patients. Myocardial ischemia/reperfusion elicits various types of cardiomyocyte death and coronary microvascular damage [17]. Therefore, cardioprotective strategies aimed at reducing infarct size that protect both cardiomyocytes and microvessels are much needed [2,3,10,14]. Here, the addition of nanocarriers charged with oxygen prior to hypoxia was protective for both cardiomyocyte-like (H9c2) and endothelial (HMEC) cell lines. When given after hypoxia, they were protective for H9c2 only, but were not damaging to HMEC. Future studies should focus first on clinically relevant animal models [18], and then on patients in severe need of adjunct cardioprotection. Hopefully, these oxygen formulations may be included in the clinical armamentarium, either in programmed intervention (programmed revascularization), or in acute ischemia/reperfusion (elective angioplasty in AMI), as they can provide protection either before or after an ischemic event. Importantly, they might protect both cardiomyocytes and microvessels, therefore boosting the potential for cardioprotection.

The release of oxygen has been the focus of extensive research, in particular for the reduction of cell death in hypoxic tissues [15,19,20,21,22,23]. The advantage of CNN is that the reduced size of the cavity allows it to host small molecules as gases. Recently, CNN in the form of a cross-linked polymer has been used, with excellent results, in the preparation of a new formulation of doxorubicin [16], a widely used anticancer drug.

The absence of oxygen, and its subsequent abrupt re-introduction, are considered the main mechanisms of ischemia/reperfusion injury. On one hand, the addition of this formulation—which releases oxygen slowly—before ischemia may represent a sort of pre-conditioning that triggers protection and builds a reservoir of oxygen that can support cell vitality during the absence of blood flow. On the other hand, the administration of this formulation immediately after ischemia may represent a sort of post-conditioning, protecting from reperfusion injury. The slow release of oxygen may limit reperfusion injury. Of note, while in pre-treatment both the 10 and 20 μg/mL concentrations were protective in both cell lines, only the higher concentration was protective in post-treatment, and in H9c2 only. Although we do not know the reasons for these differences, they open the possibility that endothelial cells are more sensitive to oxidative stress, and that different concentrations of O_2_-CNN formulation may be used in different clinical conditions. These results are interesting for the future use of CNN in clinical settings. The controlled release of oxygen could induce the activation of protective cascades, similar to conditioning, thus explaining their protective activity.

The ability to release oxygen would find numerous applications in the clinical setting, as well as at the cardiovascular level. One possible area could be the use of these oxygen carrier molecules as substitutes for hemoglobin. It is likely that, in patients with coronary disease, blood transfusion may in certain circumstances do more harm than good, even in anemic patients [24]. Therefore, substitutes for hemoglobin that overcome many of the problems associated with transfusion are much needed. Importantly, besides oxygen, the CNN may also deliver nitric oxide (NO), overcoming the putative impairment of NO effectiveness attributed to the transfused hemoglobin as well.

In conclusion, we have demonstrated here that oxygen-charged CNN has the potential to be beneficial in the context of hypoxia/re-oxygenation in vitro, limiting the cell damage in different cell lines and in a timing-specific manner. Whether CNN has beneficial effects in the context of ischemia/reperfusion in vivo remains to be ascertained. Future studies will determine the context in which these nanotools can be better exploited.

## 4. Materials and Methods

Cyclic nigerosyl-1,6-nigerose was a kind gift from HAYASHIBARA CO., LTD./NAGASE Group. All of the other chemical compounds were ACS grade and supplied by Sigma-Aldrich (St. Louis, MO, USA).

### 4.1. Preparation and Characterization of Oxygen-Loaded Cyclic Nigerosyl-Nigerose Solution

A weighted amount of cyclic nigerosyl-nigerose was dissolved in saline solution (NaCl 0.9% *w*/*v*) at a concentration of 40 mg/mL. The CNN solution was then saturated with oxygen under an oxygen purge, until reaching an oxygen concentration of up to 35 mg/L.

A second CNN sample was prepared by dissolving the CNN (40 mg/mL) in a glycerol solution (2% *w*/*v*) containing dextran-70 (6% *w*/*v*). The sample was then saturated with oxygen, as previously described. 

A glucose solution (5% *w*/*v*) containing dextran-70 (6% *w*/*v*) was used as a control, being a common plasma volume expander. This solution was saturated with oxygen.

The samples with or without oxygen were characterized in vitro by determining pH and tonicity. The pH was recorded at room temperature using an Orion 420A pH meter, while the osmolarity was measured using a Knauer K-7400 Semi-Micro Osmometer, also at room temperature. The viscosity of the samples was evaluated using a Ubbelohde capillary viscometer (Schott Gerate, Mainz, Germany).

### 4.2. Hemolytic Activity Determination

The hemolysis assay was performed to evaluate the biocompatibility of the nigerose sample.

For hemolytic activity determination, 100 µL of CNN solution at different CNN concentrations was incubated with 1 mL of blood diluted with PBS (pH 7.4, 1:4 *v*/*v*) at 37 °C for 90 min. After incubation, the samples were centrifuged at 1000 rpm for 5 min to separate the plasma. The amount of hemoglobin released due to hemolysis was determined spectrophotometrically (absorbance readout at 543 nm using a Beckman DU spectrophotometer). The hemolytic activity was calculated to compare with a negative control consisting of diluted blood without the addition of the samples. Complete hemolysis (positive control) was induced by the addition of ammonium sulphate (20% *w*/*v*).

### 4.3. In Vitro Oxygen Release Kinetics from Oxygen-Loaded Nigerose Solution

In vitro oxygen release from the CNN solution was investigated using the dialysis bag technique. The donor phase, consisting of a 3 mL sample of oxygen-loaded CNN, was placed in a dialysis bag (Spectra/Por cellulose membrane, 12 kDa) Then, it was put in 45 mL of the receiving phase, consisting of saline solution (NaCl 0.9% *w*/*v*) or cell culture medium (DMEM), whose oxygen concentration was previously reduced (up to 1 mg/L) with an N_2_ purge in order to mimic hypoxic conditions. The concentration of oxygen released from the CNN into the receiving phase was monitored for 24 h, using an oximeter (Hach HQ40d).

The in vitro oxygen release kinetics were also evaluated at 37 °C, using cell culture medium as the receiving phase.

### 4.4. Cellular Model

Cardiomyoblast (H9c2) was obtained from the American Type Culture Collection (ATCC; Manassas, VA, USA). The medium nutrient mixture of modified Dulbecco F-12 Ham (DMEM) was used to grow the H9c2 samples, in addition to 10% fetal bovine serum (FBS) and 1% (*v*/*v*) streptomycin/penicillin (Wisent Inc., Quebec, QC, Canada), at 37 °C and 5% CO_2_ [15,25,26].

Upon reaching 80% confluence the cells were detached from the flask, counted in the Burker chamber, and plated in 96-well plates with a density of 5000 cells/well. After 48 h, the H9c2 samples were subjected to dose–response analysis, growth curve, or the hypoxia/re-oxygenation (H/R) protocols. 

Human microvascular endothelial cells (HMEC) were obtained from the American Type Culture Collection. HMEC was grown in MCDB 131 Medium supplemented with 10% fetal bovine serum (FBS), 10 ng/mL of epidermal growth factor, 1 μg/mL of hydrocortisone, 2 mM glutamine, and 1% (*v*/*v*) streptomycin/penicillin, at 37 °C and 5% CO_2_ [27]. When the HMEC samples reached 80% confluence, they were seeded in 96-well plates at 5000/well and subjected to dose–response analysis, growth curve, or the H/R protocols.

### 4.5. Protocols

#### 4.5.1. Normoxic Experimental Conditions (Dose–Response Studies and Growth Curve)

To verify the toxicity level of the oxygenated (O_2_-CNN) or non-oxygenated (nitrogen (N_2_-CNN)) cyclic nigerosyl-nigerose nanosponges to either cellular model, the CNN-based formulations have been tested at different concentrations (0.2, 2, 10, and 20 μg/mL).

Therefore, the cell viability of the untreated control groups (cells with F-12 Ham DMEM, only 2% bovine serum, CTRL) was analyzed. The latter was compared with the cell viability of cells exposed to the following formulations based on O_2_-CNN and N_2_-CNN, in normoxic conditions (21% O_2_, 5% CO_2_ and 74% N_2_) (Figure 9). The growth cell curve was determined only for concentrations of 10 μg/mL of O_2_-CNN or N_2_-CNN, for different times (every 24 h for 5 days), by counting the number of cells present in the monolayer using a Burker chamber.

#### 4.5.2. Experimental Conditions of Untreated, and Pre- and Post-Treated, Cells in Hypoxia/Reoxygenation

For in-vitro hypoxia/re-oxygenation (H/R) experiments, the HMEC or H9c2 cell lines were serum-starved (FBS 2%) for 24 h.

*Pre-treatment*: Cells were pre-treated with a CNN solution saturated with O_2_ or N_2_ at four different concentrations (0.2, 2, 10, or 20 μg/mL) and exposed to hypoxia (1% O_2_, 5% CO_2_ and 94% N_2_) with an ischemic buffer (IB) containing (in mM): 137 NaCl, 12 KCl, 0.49 MgCl_2_, 0.9 CaCl_2,_ 4 HEPES, and 20 sodium lactate (pH 6.2), for 2 h. At the end of the hypoxic period, the cells were subjected to re-oxygenation (21% O_2_, 5% CO_2_ and 74% N_2_) for one hour [15].

*Post-treatment*: Cells were exposed to hypoxia (1% O_2_, 5% CO_2_ and 94% N_2_) with an IB for 2 h. At the end of the hypoxic period, the cells were treated with CNN saturated with O_2_ or N_2_ at four different concentrations (0.2, 2, 10, or 20 μg/mL) and were subjected to reoxygenation (21% O_2_, 5% CO_2_ and 74% N_2_) for one hour.

At the end of reoxygenation, the MTT test was used to assess cell viability.

### 4.6. MTT Assay

At the end of the H/R protocol, cell viability was assessed using the 3-(4,5-Dimethylthiazol-2-yl)-2,5-diphenyltetrazolium bromide (MTT) kit (10 µL/well, Sigma, St. Louis, MO, USA) as indicated by the manufactory protocol. Briefly, after 2 h incubation at 37 °C, dimethyl sulfoxide (DMSO, Sigma, St Louis, MO, USA) was added. The plates were read at 570 nm to obtain optical density values [15,25,26]. 

### 4.7. Western Blot Analysis 346 

Briefly, cells were cultivated into 6-well plates (1 × 105 cells/well) and exposed to 10 and 20 μg/mL of O_2_-CNN or N_2_-CNN under normoxic or H/R conditions as described in Figure 9. At the end of the protocol, each cell group was lysed using RIPA Lysis Buffer with proteinase and phosphatase inhibitors, and total proteins were collected for the Western blot detection. Twenty micrograms of total cell extract were separated by electrophoresis in reducing and denaturing polyacrylamide gels and transferred onto nitrocellulose membranes. After saturation with tris buffer 5% BSA, blot strips were incubated with primary antibodies. Anti-Hif-1α, anti-cleaved caspase-3 (1:1000; Abcam ab52181and Abcam ab51608, Milan, Italy), and anti-vinculin antibodies (1:1000; Sigma-Aldrich AB6039, Milan, Italy) were used as primary antibodies. The strips were incubated for 1 h at room temperature with the secondary antibodies (1:10,000; BioRad, Milan, Italy), and signals were detected using the chemiluminescent reagent LiteAblot (Euroclone, Milan, Italy). Chemidoc Touch Imaging System software (BioRad, Milan, Italy) was used for densitometric analysis [28].

### 4.8. Statistical Analysis

All values were expressed as mean ± SEM and were analyzed using the variance analysis test (ANOVA), followed by the Bonferroni post hoc test and Student’s *t* test. A value in the *p* < 0.05 range was considered statistically significant.

## 5. Conclusions

CNN has the potential to deliver oxygen, limiting cell damage and providing a benefit in the context of ischemia/reperfusion. The biological observations obtained during this first study demonstrate the ability of these carriers to release oxygen under hypoxic conditions. Future studies will determine the contexts in which these nanotools can be better exploited.

## Figures and Tables

**Figure 1 ijms-22-04208-f001:**
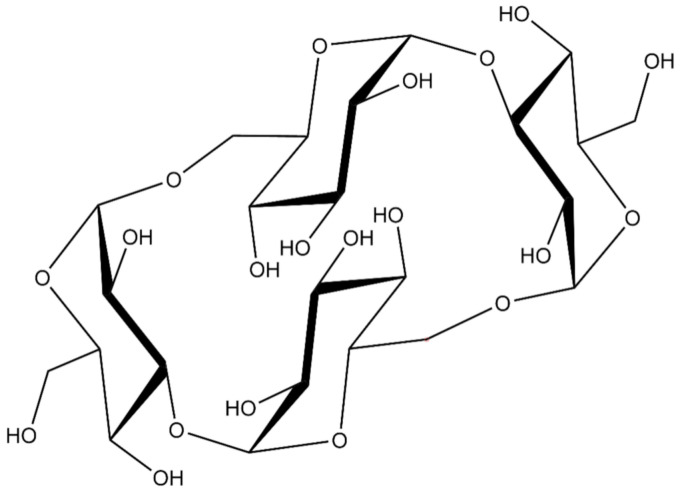
The structure of cyclic nigerosyl-nigerose (CNN).

**Figure 2 ijms-22-04208-f002:**
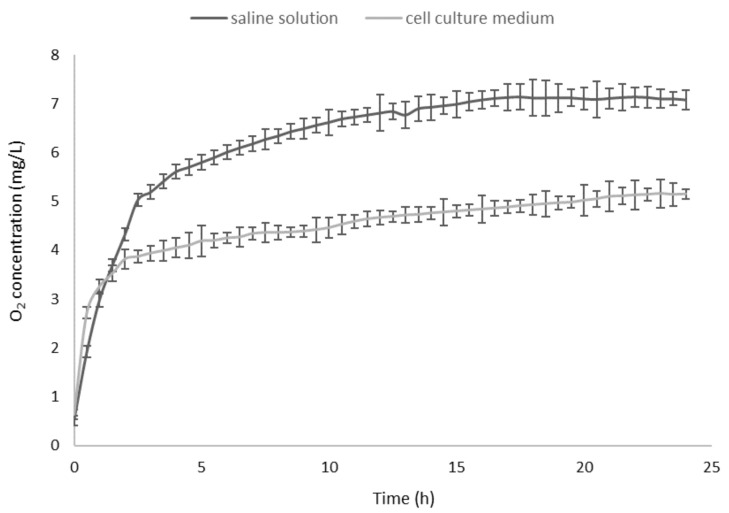
In vitro oxygen release kinetics from oxygen-loaded cyclic nigerosyl-nigerose solution (NaCl 0.9% *w*/*v*) in the two receiving phases (NaCl 0.9% *w*/*v* and cell culture medium).

**Figure 3 ijms-22-04208-f003:**
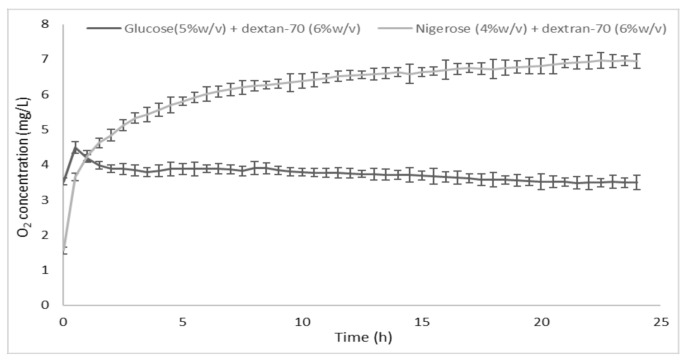
In vitro oxygen release kinetics from the oxygen-loaded cyclic nigerosyl-nigerose glycerol solution containing dextran-70 (6% *w*/*v*), compared to a 5% *w*/*v* glucose solution containing dextran-70 (6% *w*/*v*).

**Figure 4 ijms-22-04208-f004:**
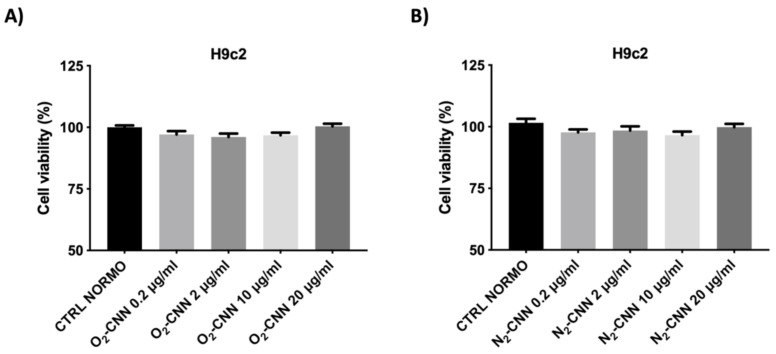
Dose–response of H9c2 in normoxic conditions. (**A**): Treatment with O_2_-CNN (0.2, 2, 10, and 20 μg/mL) compared to the untreated control group (CTRL). (**B**): Treatment with N_2_-CNN (0.2, 2, 10, and 20 μg/mL) compared to the untreated control group (CTRL). Data were expressed as mean ± SE. Data were normalized to the mean value under control conditions (CTRL NORMO) and expressed as a percentage.

**Figure 5 ijms-22-04208-f005:**
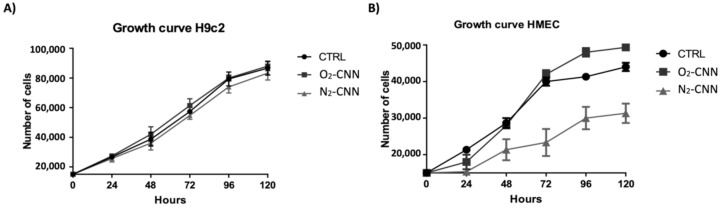
Growth curves of the two cell models. (**A**): Treatment of H9c2 with O_2_-CNN or N_2_-CNN at 10 μg/mL compared to the untreated control group (CTRL). (**B**): Treatment of HMEC with O_2_-CNN or N_2_-CNN at 10 μg/mL compared to the untreated control group (CTRL). Data were expressed as mean ± SE.

**Figure 6 ijms-22-04208-f006:**
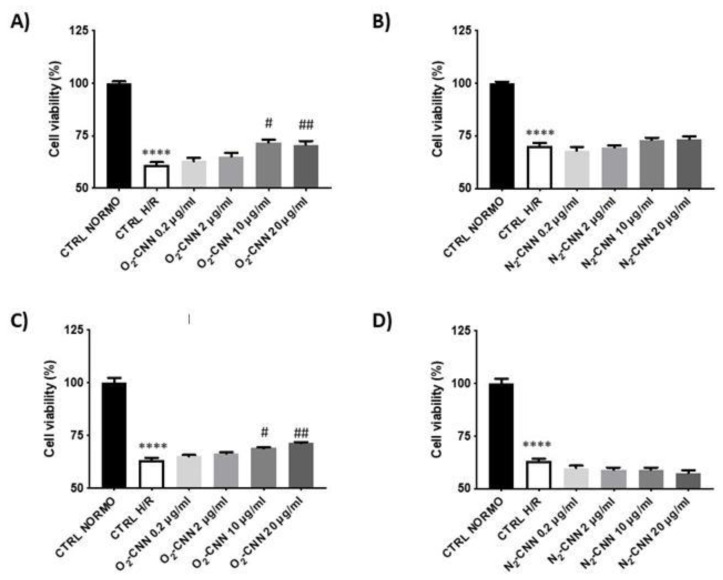
CNN pre-treated cells in H/R conditions. (**A**,**B**): O_2_- and N_2_-CNN pre-treated H9c2 in H/R conditions. (**C**,**D**): O_2_- and N_2_-CNN pre-treated HMEC in H/R conditions. **** *p* < 0.0001 vs. CTRL NORMO; ^#^
*p* < 0.01 vs. CTRL H/R; ^##^
*p* < 0.0001 vs. CTRL H/R. Data were expressed as mean ± SE. Data were normalized to the *mean value* in control conditions (CTRL NORMO) and expressed as a percentage.

**Figure 7 ijms-22-04208-f007:**
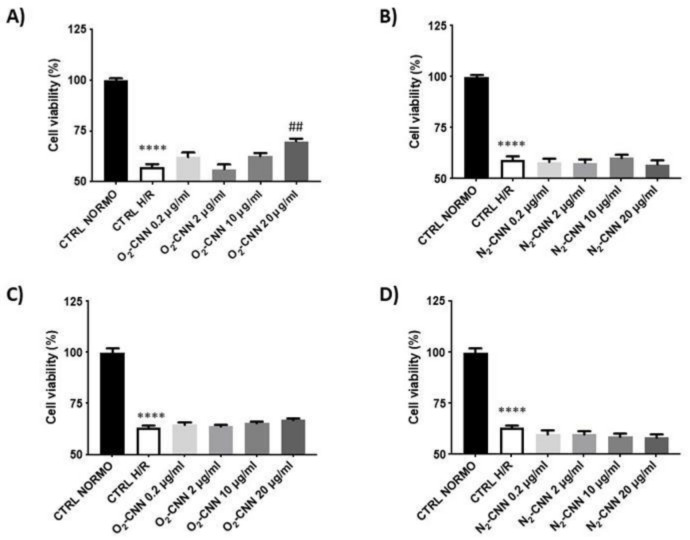
CNN post-treated cells in H/R conditions. (**A**,**B**): O_2_ and N_2_ CNN post-treated H9c2 in H/R conditions. (**C**,**D**): O_2_ and N_2_ CNN post-treated HMEC in H/R conditions. **** *p* < 0.0001 vs. CTRL NORMO; ^##^
*p* < 0.0001 vs. CTRL H/R. Data were expressed as mean ± SE. Data were normalized to the *mean value* in control conditions (CTRL NORMO) and expressed as a percentage.

**Figure 8 ijms-22-04208-f008:**
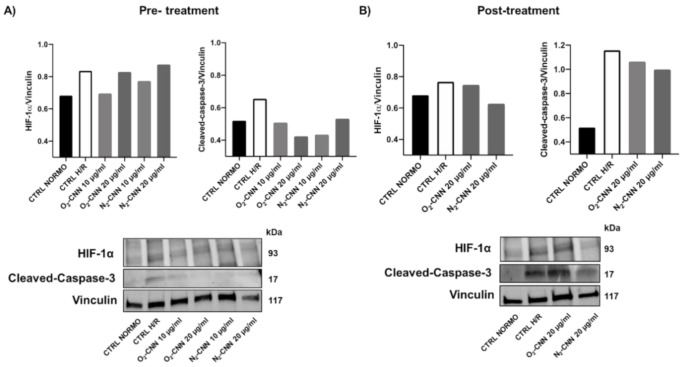
O_2_ and N_2_-CNN pre-treated H9c2 in H/R conditions. (**A**): O_2_ and N_2_-CNN post-treated H9c2 in H/R conditions. (**B**): Densitometric data and representative Western blots of HIF-1α, cleaved caspase-3, and vinculin (loading control).

**Figure 9 ijms-22-04208-f009:**
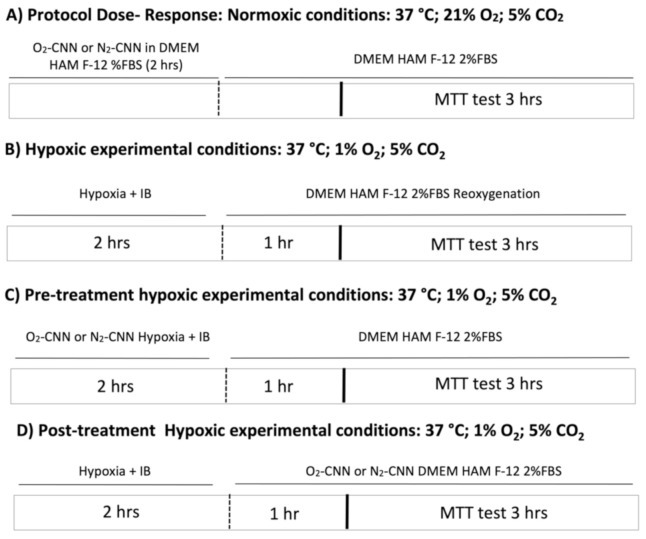
Experimental protocols: (**A**–**D**) represent different experimental conditions. 3-(4,5-dimethylthiazol-2-yl)-2,5-diphenyltetrazolium bromide (MTT) test. IB: ischemic buffer.

**Table 1 ijms-22-04208-t001:** Characteristics of oxygen-loaded cyclic nigerosyl-nigerose samples.

	Cyclic Nigerosyl-Nigerose (4% *w*/*v*) in NaCl (0.9% *w*/*v*)	Cyclic Nigerosyl-Nigerose (4% *w*/*v*) in Glycerol (2% *w*/*v*) + Dextran-70 (6% *w*/*v*)	Glucose (5% *w*/*v*) + Dextran-70 (6% *w*/*v*)
pH	6.00	6.05	5.85
Osmolarity (mOsm)	310	300	280
Viscosity (cP)	1.07	2.30	3.05

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
