# Peer review of "Cyclic Nigerosyl-Nigerose as Oxygen Nanocarrier to Protect Cellular Models from Hypoxia/Reoxygenation Injury: Implications from an In Vitro Model"

_ijms, 2021, doi:10.3390/ijms22084208_

Round 1

Reviewer 1 Report

Penna et al., have addressed the concerns highlighted in their previous version of the manuscript. 

 Major concern:

Figure 8: The information presented in the Western blots are difficult to interpret. Please add a densitometric analysis alongside. 

Minor concerns: 

Methods. Section 4.8. Western blot analysis                                                 

i) Line 348: Figure 9, in place of 99.                                                                 

ii) Please mention the sources of the antibodies individually, including their dilution;  the relevant secondary antibodies, and their source(s) as well.

Author Response

We thanks the referee for the vakuable suggestions.  We have modified as requested the Western Blot and corrected the minor concerns.

Reviewer 2 Report

Thanks for addressing all the comments

Author Response

The authors thank the referee for well considering the revision.

Reviewer 3 Report

All mistakes were corrected by authors

Author Response

The authors thank the reviewer for the positive response.